# Comparative Phytochemical, Antioxidant and Haemostatic Studies of Preparations from Selected Vegetables from *Cucurbitaceae* Family

**DOI:** 10.3390/molecules25184326

**Published:** 2020-09-21

**Authors:** Agata Rolnik, Iwona Kowalska, Agata Soluch, Anna Stochmal, Beata Olas

**Affiliations:** 1Department of General Biochemistry, Faculty of Biology and Environmental Protection, University of Łódź, 90-236 Łódź, Poland; beata.olas@biol.uni.lodz.pl; 2Department of Biochemistry and Crop Quality, Institute of Soil Science and Plant Cultivation, State Research Institute, 24-100 Puławy, Poland; ikowalska@iung.pulawy.pl (I.K.); asoluch@iung.pulawy.pl (A.S.); asf@iung.pulawy.pl (A.S.)

**Keywords:** oxidative stress, plasma, coagulation, UPLC-ESI-QTOF-MS analysis, *Cucurbitaceae* family

## Abstract

The aim of this study was to provide detailed insight into the chemical composition and activity of five cucurbit vegetable preparations (pumpkin, zucchini, cucumber, white and yellow pattypan squash), each containing various phytochemical compounds with potential use against oxidative stress induced by the hydroxyl radical donors in human plasma in vitro. We studied the antiradical capacity of vegetable preparations using the DPPH (2,2-diphenyl-1-picrylhydrazyl) method. As oxidative stress may induce changes in hemostasis, our aim included the determination of their effect on three selected hemostatic parameters of plasma, which are three coagulation times: PT (prothrombin time), APTT (activated partial thromboplastin time) and TT (thrombin time). However, none of used vegetable preparations changed APTT, PT or TT compared to the control. The phytochemical composition of the tested preparations was determined by UPLC-ESI-QTOF-MS. In our in vitro experiments, while all five tested preparations had antioxidant potential, the preparation from yellow pattypan squash showed the strongest potential. All cucurbit vegetable preparations inhibited lipid peroxidation. Only zucchini did not have an effect on protein carbonylation and only yellow pattypan squash inhibited thiol oxidation. The antioxidant activity of cucurbits appears to have triggered significant interest in multiple applications, including CVDs (cardiovascular diseases) associated with oxidative stress, which can be treated by supplementation based on these vegetables.

## 1. Introduction

Lifestyle factors, including nutrition, play an important role in the etiology and treatment of cardiovascular diseases (CVDs) [1]. Recent results have demonstrated that certain vegetables (for example onion, garlic, tomato and beetroot) and their products may act as mediators in the prevention and treatment of cardiovascular diseases by various mechanisms [1,2,3,4]. The cardioprotective actions of vegetables may include lowering of blood pressure, improving endothelial function, modifying lipid metabolism and reducing oxidative stress [2,3]. The regular intake of vegetable products rich in phenolic compounds is associated with a reduced risk of cardiovascular diseases [1,4]. However, the effect of vegetables from the *Cucurbitaceae* family on parameters of oxidative stress and hemostasis is not always well documented. In addition, the chemical components of preparations (for example extracts and fractions) isolated from these vegetables have not been adequately described.

The *Cucurbitaceae* family is a large group of crops, with 800 species. The most popular cucurbits are pumpkin, cucumber, melon and watermelon. They are cultivated and consumed in various parts of the world. Moreover, they are used in traditional medicine, for example in China, India, Mexico and Brazil [5]. Some results demonstrate that these vegetables have hepatoprotective, cardiovascular and anti-inflammatory properties [6,7]. These actions are commonly linked with chemical components. They are also rich in different phytochemicals [8].

It is known that polyphenol-rich extracts may alleviate the negative impact of oxidative stress and hemostasis [9]. Cucurbit preparations were investigated in the present work (in vitro model). This study aimed to investigate the in vitro protective effects of five cucurbit vegetable preparations: pumpkin (*Cucurbita pepo*; fruit without seeds), zucchini (*Cucurbita pepo convar. giromontina*; fruit with seeds), cucumber (*Cucumis sativus*; fruit with seeds), white pattypan squash (*Cucurbita pepo var. patisoniana*; fruit without seeds) and yellow pattypan squash (*Cucurbita pepo var. patisoniana*; fruit without seeds). We measured different parameters of oxidative stress: lipid peroxidation determined by thiobarbituric acid reactive substances (TBARS); thiol group level; protein carbonylation; oxygen radical antioxidant capacity (ORAC); and total antioxidant capacity of plasma. In addition, we also studied the antiradical capacity of preparations from cucurbit vegetables using the DPPH (2,2-diphenyl-1-picrylhydrazyl) test. As oxidative stress may induce changes in hemostasis [10], another aim of our experiments was to determine the effect of the five vegetable preparations on three selected hemostatic parameters of human plasma: activated partial thromboplastin time (APTT), prothrombin time (PT), and thrombin time (TT) in an in vitro model.

## 2. Results

### 2.1. Chemical Characteristic of Vegetable Preparations

The information on the occurrence of phytochemicals in the five cucurbit vegetable preparations with their UPLC-ESI-QTOF-MS data is shown in Table 1. A total of 36 phytochemicals were characterized, and most could be identified by comparing their retention times, MS spectra and MS/MS fragmentation and literature data [10,11,12,13,14,15]. As shown by the UPLC-ESI-QTOF-MS analysis, the five cucurbit vegetable preparations differed in their final qualitative chemical composition (Figure 1A–E), except for the two pattypan squash varieties (Figure 1B,C), whose composition was somewhat similar, with corresponding base-peak chromatograms (BPC) in the negative ionization mode of the five tested preparations. Among the identified metabolites were analytes belonging to different compound classes. The vast majority were representatives of groups such as phenylethanoid glycosides, flavonoids, fatty acids and lipids. The last class of compounds occurred in a significant amount in all the analyzed profiles and was identified as glycerophospholipids. The pumpkin (Figure 1D) and cucumber (Figure 1E) preparations showed the smallest phytochemical diversity. In these extracts, among the phenols, derivatives of kaempferol and synapic acid were identified. The extracts of zucchini (Figure 1A), white pattypan squash (Figure 1C) and yellow pattypan squash (Figure 1B) also contained phenolic compounds known as phenylethanoids, all of them occurring in the form of glycosides. Among the more known flavonoids, quercetin-3-O-rutinoside (rutin), 7-methylquercetin-3-galactoside-6″-rhamnoside-3′′′-rhamnoside (xanthorhamnin), methyl 5-methoxy-2-[(6-*O*-pentopyranosylhexopyranosyl)oxy]benzoate derivative (primulaverin), isorhamnetin 3-*O*-rutinoside (narcissin), hesperetin 7-*O*-(2″,6″-di-*O*-α-rhamnopyranosyl)-β-glucopyranoside and quercetin 3,3′-dimethyl ether 7-rutinoside were identified, with all these metabolites only being found together in the extract from zucchini. Amino acids were interpreted as two compounds, L-phenylalanine glycoside located in the extracts from pumpkin, cucumber and white pattypan squash, and l-tryptophan glycoside identified in the fractions from pumpkin, zucchini and white pattypan squash. The largest amount of compounds from the group of fatty acids was detected in the most diverse phytochemical profile of zucchini i.e., all-*cis*-6,9,12-octadecatrienoic acid (γ-linolenic acid) derivative, (9*Z*, 12*Z*)-octadeca-9,12-dienoic acid (linoleic acid) derivative and other not identified octadecadienoic acid derivatives. Two of the last compounds described were also present in the other four preparations. Many individual compounds were also identified, namely 3-(β-d-glucopyranosyloxy)-2-hydroxybenzoic acid in the preparation from yellow pattypan squash; 3,4-dihydroxyphenyl-1-methyl ester-carbamic acid in the preparation from pumpkin; cinncassiol A in the preparation from white pattypan squash; secoisolariciresinol monoglucoside in the preparations from both pattypan squashes; and, as the last identified active compound, nonanedioic acid (azelaic acid) was found in the all tested preparations.

### 2.2. Effects of Vegetable Preparations on Hemostatic Parameters of Human Plasma

The analysis of the effect on the coagulation properties of human plasma demonstrated that none of the tested cucurbit vegetable preparations (concentration range 1–50 µg/mL; incubation time 30 min) changed APTT, PT or TT compared with control (plasma without cucurbits vegetables preparations (*p* > 0.05).

### 2.3. Effects of Vegetable Preparations on Oxidative Stress Parameters

The effect of the five cucurbit vegetable preparations (concentration range 1–50 µg/mL; incubation time 30 min) on the level of the three biomarkers of oxidative stress in human plasma was studied in vitro. We observed that the tested vegetable preparations did not exert any significant effect on oxidative stress in human plasma not treated with H_2_O_2_/Fe (data not demonstrated). On the other hand, exposure of plasma to H_2_O_2_/Fe (a strong oxidant) resulted in significant protein carbonylation, oxidation of thiol groups, and enhanced levels of lipid peroxidation (Figure 2A–C). As demonstrated in Figure 2A, only four vegetable preparations (pumpkin, cucumber, white pattypan squash and yellow pattypan squash at the highest concentration of 50 µg/mL) reduced plasma protein carbonylation induced by H_2_O_2_/Fe. The best result was obtained for the pumpkin preparation (reduction of this process by more than 60% in comparison to the control positive) (Figure 2A). As shown in Figure 2B, the tested vegetable preparations had different effects on the oxidation of the protein thiols in plasma treated with H_2_O_2_/Fe. No positive effect was observed for two of the tested preparations (zucchini and white pattypan squash). On the other hand, for the yellow pattypan squash preparation, only at the highest dose (50 µg/mL) was it able to protect plasma against H_2_O_2_/Fe-induced oxidation of protein thiols. The preparation from cucumber exerted the strongest effect at lower doses (1 and 5 µg/mL) (Figure 2B). The best result was obtained for the pumpkin preparation at the concentration of 5 µg/mL (Figure 2B). Additionally, the activity of all tested vegetable preparations was not concentration-dependent for the 30 min incubation time (Figure 2B). Moreover, four of the tested preparations (zucchini, cucumber, white pattypan squash, and yellow pattypan squash) significantly (*p* < 0.05) inhibited plasma lipid peroxidation induced by H_2_O_2_/Fe starting from a dose of 1 µg/mL (Figure 2C). The pumpkin preparation showed the highest activity at higher tested concentrations (5 and 50 µg/mL) (*p* < 0.05) (Figure 2C). However, the cucurbits preparations at both concentrations (5 and 50 µg/mL) had non-significant influence on the antioxidant capacity of the ORAC plasma measures or, in concentration 50 µg/mL, the total antioxidant capacity in vitro (Figure 3A,B).

The antiradical capacities of the five vegetable preparations are also shown in Figure 3C. The order of the DPPH scavenging activity for the preparations was as follows: cucumber (0.007 ± 0.00) < pumpkin (0.052 ± 0.00) < white pattypan squash (0.064 ± 0.01) < yellow pattypan squash (0.086 ± 0.01) < zucchini (0.093 ± 0.02) (Figure 3C).

Table 2 demonstrates the comparative effects of the five tested vegetable preparations (at the highest used concentration 50 µg/mL) on selected biomarkers of oxidative stress in human plasma treated with H_2_O_2_/Fe. Table 2 gives a full picture of comparison between the effect of all selected vegetables on different biomarkers of oxidative stress in human plasma. We observed that the yellow pattypan squash preparation had a stronger antioxidant potential than the other four preparations. The yellow pattypan squash preparation inhibited oxidative stress induced by H_2_O_2_/Fe in all the in vitro tests (Table 2).

## 3. Discussion

Preparations from fruits and vegetables constitute a basis for modern phototherapy, as they contain concentrated active components, including phenolic compounds. The antioxidant or prooxidant properties of selected vegetables from the *Cucurbitaceae* family were studied in an in vitro model of human plasma exposed to oxidative stress. The stress conditions were induced by H_2_O_2_/Fe, the donor of hydroxyl radicals (one of the strongest oxidative agents) involved in the pathophysiology of various processes. Hydroxyl radicals may interact with plasma lipids and proteins. The destructive effect of OH is reflected in the increased plasma levels of oxidative stress parameters such as TBARS, oxidation of thiol groups and protein carbonylation. The levels of these biomarkers have been connected with the progress or poorer prognosis of various diseases, for example cardiovascular diseases [16]. In our in vitro experiments, all tested preparations exerted a protective action in plasma proteins and lipids against hydroxyl radicals. Therefore, our present studies indicate that the tested vegetables from the *Cucurbitaceae* family may be promising candidates for the prevention and treatment of CVDs associated with oxidative stress.

The complex chemical composition of the preparations are likely related to their ability to modulate oxidative stress, and hence reduce the risk of cardiovascular diseases. On the other hand, the preparations neither failed to induce the total antioxidant capacity of plasma nor led to higher values of ORAC. However, in the DPPH free radical scavenging activity test, all five of the preparations differed in chemical composition and demonstrated antiradical properties. The zucchini preparation showed the highest radical scavenging effect of the preparations, because its phytochemical profile was the richest in terms of the presence of flavonoid compounds (Table 1). In general, the antioxidant activity of flavonoids depends on the structure and substitution pattern of the hydroxyl groups. The essential requirement for effective radical scavenging is the 4-carbonyl group in ring C and 3′,4′-orthodihydroxy configuration in ring B. The presence of the 3-OH group or 3- and 5-OH groups, giving a catechol-like structure in ring C, is also beneficial for the antioxidant activity of flavonoids. The presence of the C2–C3 double bond configured with a 4-keto arrangement is known to be responsible for electron delocalization from ring B and increases the radical-scavenging activity. In the absence of the o-dihydroxy structure in ring B, a catechol structure in ring A can compensate for flavonoid antioxidant activity. Quercetin has a catechol structure in ring B, as well as a 2,3-double bond in conjunction with a 4-carbonyl group in ring C, allowing the delocalization of the phenoxyl radical electron to the flavonoid nucleus. The combined presence of a 3-hydroxy group with a 2,3-double bond additionally increases the resonance stabilization for electron delocalization; hence, it has a higher antioxidant value. Flavonols (e.g., quercetin, isorhamnetin) have a hydroxyl group at position 3. Kim and Lee [17] suggest a structurally important role of the 3-OH group in the chroman ring responsible for the enhancement of antioxidant activity. The antioxidant activity of the zucchini preparation is a result of rutin, isorhamnetin 3-*O*-rutinoside, hesperetin 7-*O*-(2′′,6′′-di-*O*-α-rhamnopyranosyl)-β-glucopyranoside, xanthorhamnin and quercetin 3,3′-dimethyl ether 7-rutinoside contents. The results of Iswaldi et al. [14] showed that sixteen flavonoids and other polar compounds with their derivatives were identified in the whole zucchini vegetables. The yellow pattypan squash preparation also showed a high antioxidant activity, which was significantly influenced by the very high content of benzoic acid derivatives, mainly 3-(β-d-glucopyranosyloxy)-2-hydroxybenzoic acid [14].

The results of the inorganic experimental system (DPPH test) did not quite coincide with those obtained in the biological experimental system (human plasma treated with a strong physiological inducer of oxidative stress, H_2_O_2_/Fe). However, the tested preparations showed antioxidant activities of different levels in vitro. In this biological system, in vitro, the yellow pattypan squash preparation had the strongest antioxidant properties, and inhibited oxidative stress induced by H_2_O_2_/Fe in all the tests: plasma lipid peroxidation, oxidation of thiol groups in plasma proteins and plasma protein carbonylation. In addition, it seems to be important that the preparations often demonstrated antioxidant activity even at low concentrations: 1 and 5 µg/mL. The best antioxidant activity of yellow pattypan squash may be due to the presence of a benzoic acid derivative level. The antioxidant activity of benzoic acid derivatives correlates with the presence of the phenolic group and the position of the hydroxyl group. Velika and Kron [18] discovered that derivatives with a blocked hydroxyl group showed lower antioxidant properties than derivatives with a blocked carbonyl group. The pattypan squash yellow preparation, in comparison to the other preparations, contain phenylpropanoids glycoside, which could be another reason for its antioxidant activity in plasma. Phenylpropanoid glycoside showed a strong antioxidant activity, such as inhibition of oxidation of low-density lipoprotein through free radical scavenging and metal ion chelation, which correlates with the presence of the phenylpropanoid and phenylethanoid groups in the structure. Thuan et al. [19] demonstrated that phenylpropanoid glycoside isolated from *Picria tel-ferae* can inhibit lipid peroxidation in an in vitro model using TBARS assay [19,20].

The pumpkin preparation inhibited protein carbonylation and lipid peroxidation in the plasma in the in vitro model. The pumpkin preparation was the only one to have a high content of phenolic acid, which lead to the display of antioxidant properties. Xanthopoulou et al. [21] proved a correlation between phenolic compound content and the antioxidant activity of pumpkin. The antioxidant activities of pumpkin seed extracts were determined using a DPPH free radical assay. The results showed that pumpkin seed extract demonstrated a phenol concentration-dependent antiradical activity [21]. The cucumber and white pattypan squash preparations showed similar effects to the pumpkin. Both preparations inhibited protein carbonylation and lipid peroxidation in plasma. The zucchini preparation only inhibited lipid peroxidation in plasma. Zucchini contain the highest level of flavonoids of all the tested preparations. Khennouf et al. [22] reported that flavonoids have a strong free radical scavenging ability and can inhibit xanthine oxidase activity, a source of oxygen free radicals. Flavonoids also inhibit lipids, but at a lower level than other secondary plant metabolites, such as tannins and phenolic compounds [22]. Other researchers also observed that preparations from vegetables of the *Cucurbitaceae* family may reduce oxidative stress in vitro, and in the in vivo models. For example, *Cucurbita maxima* pumpkin pectin inhibited oxidative stress induced by 2,2′-azobis(2-methylpropionamide) dihydrochloride in cell cultures (cell lines HT-29, human colon adenocarcinoma), and MDCK1 (canine kidney epithelium). Oxidative stress was measured by the production of intracellular reactive species [23]. In another in vitro model, Shayesteh et al. [24] also demonstrated the protective effects of pumpkin fruit extract against oxidative stress. Cumene hydroperoxide and glyoxal were used as inductors of oxidative stress in freshly isolated rat hepatocytes. The tested extract (50 µg/mL) reduced oxidative stress as measured by various parameters, including lipid peroxidation, reactive oxygen species production and glutathione depletion. The same activity was observed by Bahramsoltani et al. [25]. Moreover, the results of Abarikwu et al. [26] demonstrated that fluted pumpkin seeds (200 mg/kg body wt.) protect against busulfon-induced oxidative stress in adult mice. The effect of ethanolic extract of fluted pumpkin seeds was investigated after 40 days of oral administration. Ghahremanloo et al. [27] have also observed that pumpkin extract ameliorates oxidative stress in obese rats, leading to decreased cardiovascular disease risk in obesity. Three groups of obese rats received hydroalcoholic extract of pumpkin as one daily dose of 100, 200 and 400 mg/kg, respectively. At the end of six weeks, the parameters of oxidative stress were measured [27].

Oxidative stress may alter the coagulation process, and this may lead to the development of CVDs [16]. However, it is not known whether the tested preparations are associated with the modulation coagulation process, or whether they have an antithrombotic activity. For the first time, in our present study on these five preparations, we found that they did not change the coagulation system, and did not show anticoagulant or procoagulant potential in the in vitro model. Therefore, we suggest that the preparations may be a source of antioxidants that do not incur the risk of bleeding or thrombosis [16].

## 4. Materials and Methods

### 4.1. Chemicals

Methanol and acetonitrile, HPLC grade, were purchased from Merck (Darmstadt, Germany). Formic acid, LC-MS grade, was purchased from Sigma-Aldrich, (St. Louis, MO, USA). Ultrapure water was obtained in-house with a purification system (Milli-Q-Simplicity-185, Millipore Corp.). Dimethylsulfoxide (DMSO), thiobarbituric acid (TBA) and H_2_O_2_ were purchased from Sigma (St. Louis, MO, USA). Other reagents were of analytical grade and were provided by commercial suppliers, including POCh, (Poland), Acros (Poland) and Chempur (Poland).

### 4.2. Plant Material

Five cucurbit vegetable types were selected, and subjected to freeze-drying (CHRIST Gamma 2-16 LSC Freeze Dryers, Osterode am Harz, Germany): pumpkin (*Cucurbita pepo* L., fruit without seeds); zucchini (*Cucurbita pepo* L. *convar. Giromontina*, fruit with seeds); cucumber (*Cucumis sativus* L., fruit with seeds); white pattypan squash (*Cucurbita pepo* L. var. *patisoniana*, fruit without seeds) and yellow pattypan squash (*Cucurbita pepo* L. var. *patisoniana*, fruit without seeds). The plant material was stored at the Department of Biochemistry and Crop Quality of the Institute in Puławy, Poland.

### 4.3. Extraction and Preparation of Vegetable Preparations

Plant materials (1 mg of each of the freeze-dried vegetables) were extracted using an automatic extractor, Dionex ASE 200 Accelerated Solvent Extraction System. The extraction process conditions were as follows: extraction solvent: 80% methanol, solvent pressure: 1500 psi, extraction cell temperature: 40 °C, extraction cycles: 3. The extracts were evaporation dried under reduced pressure, at 40 °C (Heidolph Hei-Vap Advantage, rotary evaporator). The five extracts were purified, mainly from sugars, by solid phase extraction (SPE). Specialized metabolite fractions were eluted from Oasis Extraction Cartridges (Waters, MA, USA) with 6 mL of 85% methanol. The evaporated eluate was dissolved in 70% MeOH. Next, the samples were cleaned in an ultrasonic bath for 5 min at 30 °C (SONOREX DIGITEC DT 510 H, Bandelin, Germany) and centrifuged for 5 min at 10,000 rpm at 20 °C (laboratory centrifuge: Polygen Sigma 3-16 KL, Sigma, Germany). Finally, 150 μL of the supernatant from each sample was subjected to HPLC-MS analysis.

### 4.4. Phytochemical Profiling

The quantitative analysis of each preparation was carried out by ultra-high resolution mass spectrometry (UHRMS) using a Dionex UltiMate 3000RS (Thermo Scientific, Darmstadt, Germany) system with a charged aerosol detector (CAD) interfaced with a high-resolution quadrupole time-of-flight mass spectrometer (HR/Q-TOF/MS, Impact II, Bruker Daltonik GmbH, Bremen, Germany). The chromatographic separation was performed on an Acquity UPLC BEH C18 column (100 × 2.1 mm, 1.7 μm, Waters, Manchester, UK), with the column temperature maintained at 60 °C. The mobile phases were acidified (0.1% formic acid) water (solvent A) and acidified (0.1% formic acid) acetonitrile (solvent B). The chromatographic method consisted of the following linear gradient: 7% B from 0 to 0.5 min and then the concentration of B was increased to 80% from 0.5 to 26 min. The sample injection volume was 1.0 μL and the flow rate was set at 500 μL/min. Compounds were analyzed based on data from the mass spectra. Electrospray ionization (ESI) was performed in negative ion mode. The mass scan range was set at 50–2000 *m*/*z*. Ion source parameters were as follows: capillary voltage 3.0 kV, collision energy20.0 eV, dry gas 6.0 L/min and dry temperature 200 °C. Data acquisition and processing were performed using DataAnalysis 4.3 (Bruker Daltonik GmbH, Bremen, Germany). Determination of molecular formula was carried out by mass accuracy, adduct, and fragment information using SmartFormula.

### 4.5. Stock Solutions of Vegetable Preparation

The stock solutions of vegetable preparations, used in the tests for biological activity, were made in 50% DMSO. The final concentration of DMSO in the samples (human plasma) was lower than 0.05% and its effects were determined in all experiments.

### 4.6. Human Plasma Isolation

Human blood or plasma were obtained from six regular donors (non-smoking men and women) to a blood bank (Lodz, Poland) and a medical center (Lodz, Poland). Blood was collected as a CPD solution (citrate/phosphate/dextrose; 9:1; *v*/*v* blood/CPD) or CPDA solution (citrate/phosphate/dextrose/adenine; 8.5:1; *v*/*v*; blood/CPDA). Donors had not taken any medication or addictive substances (including tobacco, alcohol and antioxidant supplementation) for at least two week before a donation. Our analysis of the blood samples was performed under the guidelines of the Helsinki Declaration for Human Research, and approved by the Committee on the Ethics of Research in Human Experimentation at the University of Lodz (resolution ref. 8/KBBN-UŁ/III/2018). The plasma was isolated by differential centrifuging as described earlier [28]. The plasma was incubated (30 min, at 37 °C; for hemostatic parameters) with vegetable preparations at the final concentrations of 1–50 µg/mL. Human plasma was also pre-incubated (5 min, at 37 °C; for parameters of oxidative stress) with vegetable preparations at the final concentrations of 1–50 µg/mL, and then treated with 4.7 mM H_2_O_2_/3.8 mM Fe_2_SO_4_/2.5 mM EDTA (25 min, at 37 °C).

The protein concentration, determined by measuring absorbance at 280 nm according to the procedure of Whitaker and Granum [29], was measured using Bradford protein assay [30].

### 4.7. Markers of Oxidative Stress

#### 4.7.1. Lipid Peroxidation Measurement

Lipid peroxidation was quantified by measuring the concentration of TBARS, according to the method described by Wachowicz [31] and Bartosz [32]. After 30 min of incubation with a preparation from Cucurbitaceae vegetables at the final concentrations of 1–50 µg/mL, the samples were mixed with an equal volume of cold 15% (*v*/*v*) trichloroacetic acid (C_2_HCl_3_O_2_) in 0.25M HCl and 0.37% (*v*/*v*) TBA in 0.25 M HCl, and then immersed in a boiling water bath for 15 min. After cooling, the absorbance was measured at 535 nm using the SPECTROstar Nano Microplate Reader (BMG LABTECH, Ortenberg, Germany). The TBARS concentration was calculated using the molar extinction coefficient (ε = 156,000 M^−1^ cm^−1^) and was expressed as nmol/mL of plasma.

#### 4.7.2. Carbonyl Group Measurement

The carbonyl groups were determined in plasma proteins according to Levine et al. [33] and Bartosz [32]. The absorbance measurement (at λ = 375 nm) was performed using the SPECTROstar Nano Microplate Reader (BMG LABTECH, Ortenberg, Germany). The carbonyl group concentration was calculated using a molar extinction coefficient (ε = 22,000 M^−1^ cm^−1^) and was expressed as nmol/mg of plasma protein.

#### 4.7.3. Thiol Group Measurement

The thiol group content was measured spectrophotometrically (absorbance at λ = 412 nm), using a SPECTROstar Nano Microplate Reader (BMG LABTECH, Ortenberg, Germany), with 5,5′-dithio-bis-(2-nitrobenzoic acid) (Ellman’s reagent) according to the method described by Ando and Steiner [34,35] and Bartosz [32]. The thiol group concentration was calculated using a molar extinction coefficient (ε = 13,600 M^−1^ cm^−1^) and was expressed as nmol/mL of plasma protein.

#### 4.7.4. TLC-DPPH• Test

The free radical scavenging potential of the pumpkin, zucchini, cucumber, white and yellow pattypan squash preparations was assessed by means of a simple benchtop TLC-DPPH• bioassay with ImageJ program. This method, with small modifications, has been found suitable for the analysis of complex samples, as proved in our previous publications [36,37,38].

Preparations (5 mg/mL) and standard compounds, chlorogenic acid ((1*S*,3*R*,4*R*,5*R*)-3-{[(2*E*)-3-(3,4-dihydroxyphenyl)prop-2-enoyl]oxy}-1,4,5-trihydroxycyclohexanecarboxylic acid) and rutin (2-(3,4-dihydroxyphenyl)-5,7-dihydroxy-3-[α-L-rhamnopyranosyl-(1→6)-β-D-glucopyranosyloxy]-4*H*-chromen-4-one), 1 mg/mL, were prepared and applied to aluminum-backed silica gel (60 F254, Merck) chromatographic plates, with an 8 mm gap between them, and at 10 mm from both the left and low edges, using a micropipette with a scale. The plates were developed in vertical chambers pre-saturated for 15 min with the optimized mobile phase: acetonitrile:chloroform:water:formic acid (80:10:10:5, *v*/*v*/*v*/*v*). The plates were developed at a distance of 90 mm and dried in a hood for 30 min before derivatization. Next, the TLC plates were immersed for 5 s in a freshly prepared 0.2% (*w*/*v*) methanolic DPPH• solution. After removing the DPPH• excess, the plates were stored in the dark for 30 min and then scanned by means of a flatbed scanner. Compounds with the ability to scavenge free radicals emerged as a yellow band against a purple background. The test was performed in triplicate.

#### 4.7.5. Image Processing Procedure

The results of the TLC–DPPH• test were documented by flat-bed scanning, saved in the form of jpg image files and further processed by means of an open source and free program, ImageJ, developed at the National Institute of Health in the USA.

For the DPPH• staining, the results change over time, and therefore it is crucial to precisely define the time that elapses between immersion and documentation. The results were documented every 10 min for an hour, and after the comparison it was decided to process the images taken 30 min after staining. Subsequently the images chosen were processed by means of the ImageJ program, with the use of a modified procedure following Olech et al. [39]. In summary, the color images were converted to 8-bit type images (Image/Type/8-bit). These images were denoised by applying the following steps: Process/Filters/Median/Radius-20 pixels. The baseline drift was removed (Process/FFT/Bandpass Filter/Filter large structures down to-120 pixels; filter small structures up to-0 pixels). The images processed in this way were then inverted (Edit/Invert). In order to change the videoscan images into chromatograms, resembling those obtained in high-performance liquid chromatography (HPLC), a rectangular selection tool was used to outline the tracks. The line profile plots were obtained in the same way as described in the original procedure [39]. The areas under the common peaks were measured and compared with the area obtained for chlorogenic acid and rutin, a compound with a recognized free radical scavenging potential.

#### 4.7.6. ORAC Assay

ORAC is based on a hydrogen atom transfer (HAT) process, with oxidation of a fluorescent probe by peroxyl radicals. The role of the antioxidant in the assay is to block peroxyl radical oxidation of the fluorescent probe until the whole antioxidant activity in the sample is complete. The sample antioxidant activity is associated with the fluorescence decay curve, which is represented as the area under the curve (AUC), used to measure total peroxyl radical antioxidant activity and compare to the antioxidant standard curve of Trolox, a water-soluble tocopherol analogue. The Trolox curve was prepared on the same plate as all samples and according to the instructions supplied in the kit. The procedure for the ORAC assay was performed on plasma according to the instructions supplied with the Oxygen Radical Antioxidant Capacity (ORAC) Assay kit from CELL BIOLABS, INC (San Diego, USA). The kit included a 96-well microtiter plate with clear bottom black plate, fluorescein probe 100×, free radical initiator, antioxidant standard (Trolox™), and assay diluent (4×). The samples at concentration 5 and 50 µg/mL were dissolved in the ratio 1:100 [40].

#### 4.7.7. Total Antioxidant Capacity of Plasma

Total antioxidant capacity of plasma (with a preparation from *Cucurbitaceae*) can be measured by the antioxidant ability of metmyoglobin to inhibit the oxidation of 2,2′-azino-di-[3-etyhylbenthiazoline sulphonated] (ATBS) in the sample. The amount of ABTS can be monitored by measuring the absorbance at 750 nm. The antioxidant in the sample causes suppression of the absorbance to a degree proportional to its concentration. The antioxidant capacity of the sample is compared with that of Trolox, a water-soluble tocopherol analogue. The Trolox curve was prepared on the same plate as all samples and according to instructions supplied in kit. The procedure for total antioxidant capacity was performed on plasma according to the instructions supplied with the Antioxidant Assay KIT from Cayman Chemical (Ann Arbor, MI, USA). The kit included Antioxidant Assay Buffer (10×), Antioxidant Assay Chromogen, Antioxidant Assay Metmyoglobin, Antioxidant Assay Trolox, Antioxidant Assay Hydrogen Peroxide, and a 96-well solid plate. The samples at concentration 50 µg/mL were dissolved 1:20 [41].

### 4.8. Parameters of Coagulation

#### 4.8.1. Measurement of Prothrombin Time (PT)

The PT was determined coagulometrically using an Optic Coagulation Analyser, model K-3002 (Kselmed, Grudziadz, Poland), according to the method described by Malinowska et al. [42]. Briefly, after 30 min of treatment with a *Cucurbitaceae* preparation, the human plasma (50 μL) was incubated for 2 min at 37 °C and then, directly before measurement, 100 μL of Dia-PT liquid (commercial preparation: Kselmed, Grudziadz, Poland) was added.

#### 4.8.2. Measurement of Thrombin Time (TT)

The TT was determined coagulometrically using an Optic Coagulation Analyser, model K-3002 (Kselmed, Grudziadz, Poland), according to the method described by Malinowska et al. [42]. Briefly, after treatment with the *Cucurbitaceae* preparation, the human plasma (50 μL) was incubated for 1 min at 37 °C and then, directly before measurement, 100 μL of thrombin was added (final concentration was 5 U/mL).

#### 4.8.3. Measurement of Activated Partial Thromboplastin Time (APTT)

The APTT was determined coagulometrically using an Optic Coagulation Analyser, model K-3002 (Kselmed, Grudziadz, Poland), according to the method described by Malinowska et al. [42]. Briefly, after treatment with the *Cucurbitaceae* preparation, the human plasma (50 μL) was incubated with 50 μL of Dia-PTT liquid (commercial thromboplastin: Kselmed, Grudziadz, Poland) for 3 min at 37 °C and then, directly before measurement, 50 μL of 25 mM CaCl2 was added.

### 4.9. Data Analysis

Several tests were used to carry out the statistical analysis. All the values in this study were expressed as mean ± SE. The results were first evaluated for normality with the Kołmogorowa–Smirnowa test and equality of variance with the Levine test. Statistically significant differences were assessed by applying the ANOVA test (significance level was *p* < 0.05), followed by a Tukey multiple comparisons test or Kruskal–Wallis test.

## 5. Conclusions

The present paper is the first detailed study of these five preparations from selected vegetables from the *Cucurbitaceae* family and provides new insights into their phytochemical composition and biological activity (using human plasma). In this work, the UHPLC-ESI-QTOF-MS system was used to identify different compounds in the tested preparations. The results revealed that these preparations have various bioactive compounds with antioxidant activities for use in the prophylaxis and treatment of diseases involving oxidative stress, including CVDs. Although antioxidant properties were demonstrated in an in vitro model, the real effect of these extracts should be verified in an in vivo model.

## Figures and Tables

**Figure 1 molecules-25-04326-f001:**
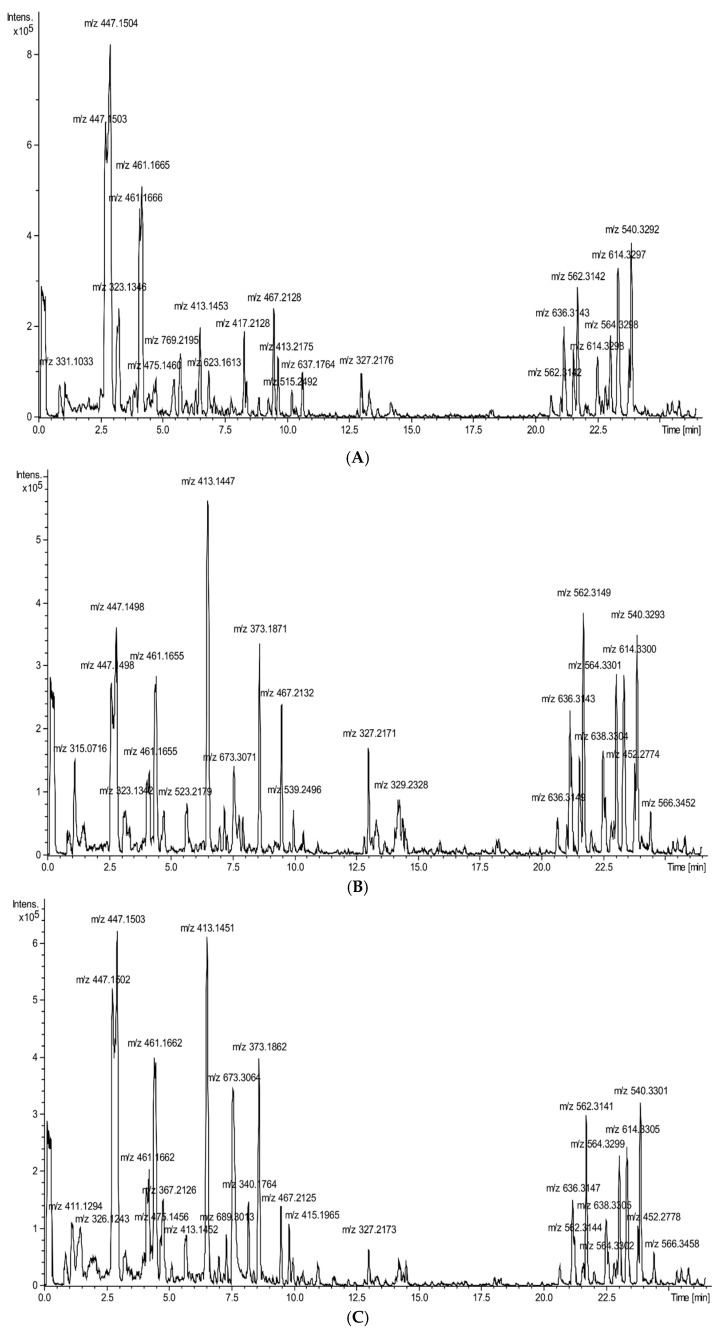
The base-peak chromatogram (BPC) of the five cucurbit vegetable preparations, obtained using high resolution UHPLC-ESI-QTOF-MS in negative ionization mode: preparation from zucchini (**A**), preparation from yellow pattypan squash (**B**), preparation from white pattypan squash (**C**), preparation from pumpkin (**D**), and preparation from cucumber (**E**).

**Figure 2 molecules-25-04326-f002:**
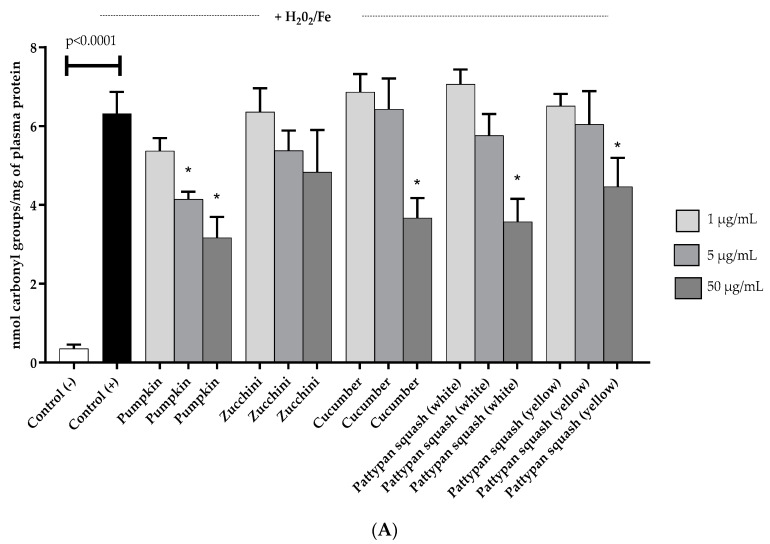
Effects of the five cucurbit vegetable preparations (concentration range 1–50 µg/mL, pre-incubation time 5 min) on the oxidative damages of plasma protein–protein carbonylation, in plasma treated with H_2_O_2_/Fe (incubation time-25 min) (**A**); on the oxidative damages of plasma proteins, the level of thiol groups in plasma treated with H_2_O_2_/Fe (incubation time 25 min) (**B**); and on lipid peroxidation in plasma treated with H_2_O_2_/Fe (incubation time 25 min) (**C**). Results are given as mean ± SE (*n* = 6). Control negative refers to plasma not treated with H_2_O_2_/Fe, whereas control positive to plasma treated with H_2_O_2_/Fe. One-way ANOVA followed by a multicomparison Tukey test and Kruskal–Wallis test: * *p* < 0.05, compared with positive control (treated with H_2_O_2_/Fe).

**Figure 3 molecules-25-04326-f003:**
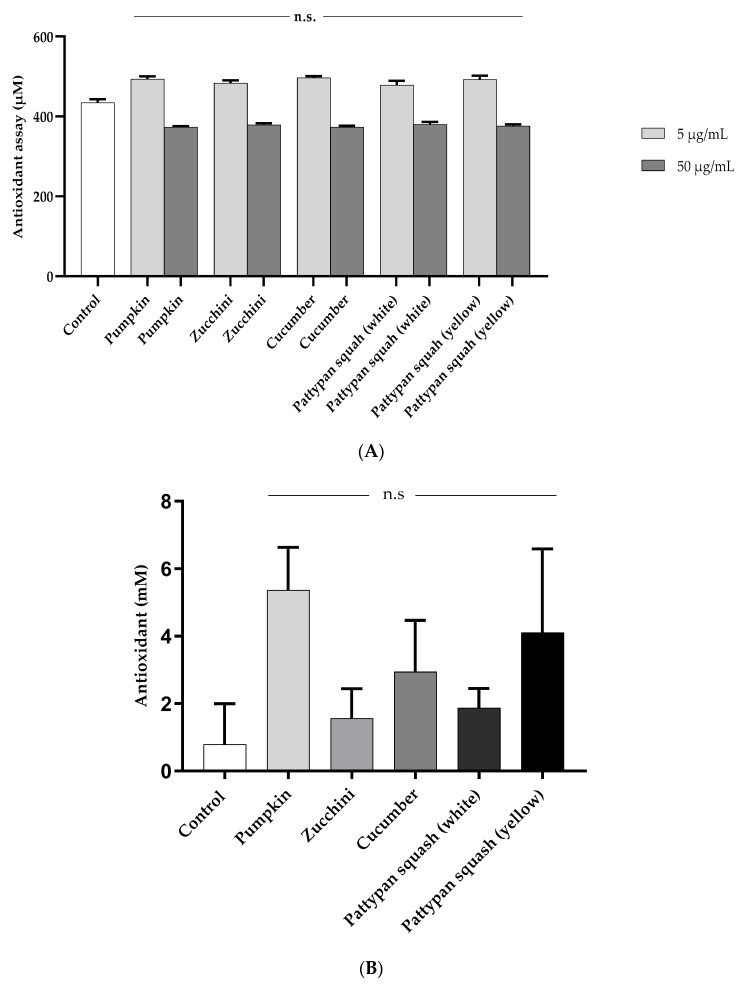
Effects of the five cucurbit vegetable preparations (incubation time 30 min) on the plasma antioxidant capacity, measured as ORAC (oxygen radical antioxidant capacity) (**A**); on total antioxidant capacity of plasma (**B**). Results are given as mean ± SE (*n* = 5). One-way ANOVA followed by a multicomparison Tukey test and Kruskal–Wallis test, compared with control. The use of TLC-DPPH• assay for the detection of antioxidant activity of preparations from: 1-pumpkin, 2-zucchini, 3-cucumber, 4-white pattypan squash, 5-yellow pattypan squash. Standards: chlorogenic acid (CGA) and rutin (R) (**C**).

**Table 1 molecules-25-04326-t001:** UPLC-ESI-QTOF-MS data of identified compounds and their presence (+) in five cucurbit vegetable preparations.

Nr	Rt (min)	Identified Compound	Compound Class	[M-H], m/z, ESI Neg.	Major MS-MS Fragments	Pumpkin	Cucumber	Zucchini	White Pattypan Squash	Yellow Pattypan Squash
1	1.14	3-(*β*-D-glucopyranosyloxy)-2-hydroxybenzoic acid	benzoic acid derivative	315.071649	152, 315	-	-	-	-	+
2	1.47	fructosyl l-phenylalanine	amino acid	326.124313	164, 236, 326	+	+	-	+	+
3	1.87	l-tryptophan glycoside	amino acid	365.134743	116, 203, 275, 365	+	-	-	-	-
4	2.45	salicylic acid *O*-glycoside	phenolic acid	299.077241	137, 299	-	+	-	-	-
5	3.87	zizybeoside I	phenylethanoid glycoside	431.155861	147, 431	-	-	+	-	-
6	4.23	forsythoside E (isomer I)	phenylethanoid glycoside	461.166185	147, 309, 461	-	-	+	+	+
7	4.35	cinncassiol A	diterpenoid	381.191905	289, 381	-	-	-	+	-
8	4.39	sinapic acid hexoside	phenolic acid	431.192317	223, 385, 431	+	-	-	-	-
9	4.43	hydrangeifolin I	phenylpropanoid glycoside	415.160971	269, 415, 461	-	-	-	+	+
10	4.55	shimaurinoside B	megastigmane glycosides	381.176621	249, 381, 427	+	-	-	-	-
11	4.65	kaempferol derivative	flavonoid	450.117527	145, 285, 450	-	+	-	-	-
12	4.68	primulaverin derivative	flavonoid	475.146049	133, 295, 323, 475	-	-	+	-	-
13	4.75	adenostemmoic acid C	diterpenoid	367.212612	287, 303, 367	-	-	-	-	+
14	5.50	rutin	flavonoid	609.146766	301, 609	-	-	+	-	-
15	5.69	secoisolariciresinol monoglucoside	lignan	523.217892	165, 361, 523	-	-	-	+	+
16	5.75	xanthorhamnin	flavonoid	769.219503	299, 314, 769	-	-	+	-	-
17	6.54	unidentified	iridoid glycoside	413.145347	269, 311, 351, 413	-	-	+	+	+
18	6.88	isorhamnetin 3-*O*-rutinoside	flavonoid	623.161334	299, 315, 623	-	-	+	-	-
19	7.21	azelaic acid	dicarboxylic acid	187.097798	125, 187	+	+	-	-	+
20	8.41	hesperetin 7-*O*-(2′′,6′′-di-*O*-*α*-rhamnopyranosyl)-*β*-glucopyranoside	flavonoid	739.244676	295, 471, 559, 739	-	-	+	-	-
21	9.68	octadecadienoic acid derivative	fatty acid	413.217479	209, 371, 413	-	-	+	-	-
22	10.66	quercetin 3,3′-dimethyl ether 7-rutinoside	flavonoid	637.176449	299, 313, 329, 637	-	-	+	-	-
23	13.04	octadecadienoic acid derivative	fatty acid	327.218013	211, 327	+	-	+	+	+
24	14.17	octadecadienoic acid derivative	fatty acid	329.233674	211, 329	-	+	+	+	+
25	20.68	glycerophospholipid	lipid	636.316333	277, 474, 636	-	+	+	+	+
26	21.19	glycerophospholipid	lipid	636.315949	277, 474, 636	+	+	+	+	+
27	21.26	glycerophospholipid	lipid	562.315676	277, 505, 562	+	+	+	+	+
28	21.57	γ-linolenic acid derivative	fatty acid	721.364358	277, 397, 721	-	-	+	-	+
29	22.54	glycerophospholipid	lipid	638.331427	152, 279, 476, 638	+	+	+	+	+
30	22.63	glycerophospholipid	lipid	564.331348	279, 504, 564	+	+	+	+	+
31	23.03	γ-linolenic acid derivative	fatty acid	559.311446	277, 559	-	-	+	-	-
32	23.06	linoleic acid derivative	fatty acid	564.329757	279, 504, 564	+	+	+	+	+
33	23.27	glycerophospholipid	lipid	614.330267	255, 452, 614	+	+	+	+	+
34	23.82	glycerophospholipid	lipid	452.277974	255, 452	-	-	+	+	+
35	23.89	glycerophospholipid	lipid	540.330582	255, 480, 540	+	+	+	+	+
36	24.46	glycerophospholipid	lipid	566.346313	281, 506, 566	+	+	-	+	+

**Table 2 molecules-25-04326-t002:** Comparative effects of the five cucurbit vegetable preparations on oxidative stress in plasma treated with H_2_O_2_/Fe (tested concentration 50 µg/mL).

Preparation	Protein Carbonylation	Thiol Oxidation	Lipid Peroxidation
from fruit with seeds			
Zucchini	No effect	No effect	Positive action-inhibition of this process (anti-oxidative potential)
Cucumber	Positive action-inhibition of this process (anti-oxidative potential)	No effect	Positive action-inhibition of this process (anti-oxidative potential)
from fruit without seeds			
Pumpkin	Positive action-inhibition of this process (anti-oxidative potential)	No effect	Positive action-inhibition of this process (anti-oxidative potential)
White pattypan squash	Positive action-inhibition of this process (anti-oxidative potential)	No effect	Positive action-inhibition of this process (anti-oxidative potential)
Yellow pattypan squash	Positive action-inhibition of this process (anti-oxidative potential)	Positive action-inhibition of this process (anti-oxidative potential)	Positive action-inhibition of this process (anti-oxidative potential)

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
