# Peer review of "Comparative Phytochemical, Antioxidant and Haemostatic Studies of Preparations from Selected Vegetables from Cucurbitaceae Family"

_molecules, 2020, doi:10.3390/molecules25184326_

Round 1

Reviewer 1 Report

Reviewer comments:

Article: Comparative Phytochemical, Antioxidant and Haemostatic Studies of Preparations from Selected Vegetables from Cucurbitaceae Family

The presente study demonstrates the identification of chemical compounds (UPLC-ESI-QTOF-MS) and antioxidante potential (oxidative stress induced by the hydroxyl radical donors in human plasma in vitro and  antiradical capacity using the DPPH method), as well as the anticoagulante effect (APTT, TT and PT tests, in vitro) of five cucurbit vegetable preparations (pumpkin, zucchini, cucumber, white and yellow pattypan squash). The five samples (preparations) demonstrates antioxidant potential with different patterns of inhibition, related to their chemical characteristics. The preparation from yellow pattypan squash showed the strongest one. However, none of the samples showed an effect on the coagulation system. Based on the composition and antioxidant effect of the studied samples, it is suggested that they can be applications, as supplement, in the treatment of diseases associated with increased oxidative stress, such as CVDs.

  1. Abstract: include more data of the results obtained;
  2. Abstract: Line 19: “haemostatic parameters of plasma” – what parameters? It was not commented on the results obtained.
  3. Line 52: delete extra parentheses;
  4. Line 311: padronizar separação das unidades
  5. Line 313: include centrifuge temperature;
  6. Line 314: what analysis are you referring to?
  7. Line 351: delete extra parentheses;
  8. Line 350-352: “The protein concentration, determined by measuring absorbance at 280 nm (in the tested samples), was calculated according to the procedure of Whitaker and Granum [12] and was measure using Bradford protein assay.” - Include the results (proteins) and reference of the Bradford method;
  9. Line 423: standardize reference [22] or (Hu et al. 2020).
  10. Line 436: standardize reference [23] or (Mesa et al. 2015);
  11. Line 577: include the number of the reference;
  12. In the methodology or results add the yields obtained from each extraction;
  13. How much weight was used for each fruit? Include detail in methods;
  14. Fig 4B: what was the concentration used of the extracts?
  15. Remove additional information from the top of the figure (Fig 1A-E);
  16. Line 212: “highest content of flavonoids” - it was quantified?
  17. Include references in some paragraphs of the discussion;
  18. In vitro/in vitro: standardize word;
  19. Line 241: “benzoic acid derivative level, around 260 mg on 1 g of dry” - this result was described? What about the values ​​obtained for the other samples?
  20. Line 266: remove color markup;
  21. It would be important to include the quantitative values ​​(dry weight) of the main compounds present in the samples;
  22. I suggest performing in vitro assays on platelet aggregation. Some studies have demonstrated a relationship between oxidative stress and platelets/thrombosis (e.g.: Oxidative Stress and Platelets. Arteriosclerosis, Thrombosis, and Vascular Biology, Volume 28, Issue 3, 1 March 2008, Pages s11-s16). In addition, the antiplatelet / antithrombotic effect of polyphenols has already been demonstrated;
  23. Line 193: “including extracts and fractions” - which fractions?
  24. Suggestion: the results (figures) of the effect on blood clotting can be suppressed and the results presented only within the text.

Author Response

Dear Reviewer

Thank you for heplfull comments  and suggestions on our manuscript. I would like to respond for comments below:

Article: Comparative Phytochemical, Antioxidant and Haemostatic Studies of Preparations from Selected Vegetables from Cucurbitaceae Family

The presente study demonstrates the identification of chemical compounds (UPLC-ESI-QTOF-MS) and antioxidante potential (oxidative stress induced by the hydroxyl radical donors in human plasma in vitro and  antiradical capacity using the DPPH method), as well as the anticoagulante effect (APTT, TT and PT tests, in vitro) of five cucurbit vegetable preparations (pumpkin, zucchini, cucumber, white and yellow pattypan squash). The five samples (preparations) demonstrates antioxidant potential with different patterns of inhibition, related to their chemical characteristics. The preparation from yellow pattypan squash showed the strongest one. However, none of the samples showed an effect on the coagulation system. Based on the composition and antioxidant effect of the studied samples, it is suggested that they can be applications, as supplement, in the treatment of diseases associated with increased oxidative stress, such as CVDs.

  1. Abstract: include more data of the results obtained;

Response: We added more date of the results: “However none of used vegetables preparations changed APTT, PT or TT compared with control. All cucurbits vegetables preparations inhibited lipid peroxidation. Only zucchini did not have effect on protein carbonylation and only yellow pattypan squash inhibited thiol oxidation”.

  1. Abstract: Line 19: “haemostatic parameters of plasma” – what parameters? It was not commented on the results obtained.

Response: Those heamostatic parameters were three coagulation times PT, APTT and TT.

  1. Line 52: delete extra parentheses

Response: We deleted the extra parentheses.

  1. Line 311: padronizar separação das unidades

Response: Sorry, but we did not understand. Can you translated to English? I’m not sure in which language that sentence is.

  1. Line 313: include centrifuge temperature;

Response: Temperature 20°C has been added (line 313).

  1. Line 314: what analysis are you referring to?

Response: I referring to a HPLC-MS analysis (line 314).

  1. Line 351: delete extra parentheses;

Response: We deleted the extra parentheses.

  1. Line 350-352: “The protein concentration, determined by measuring absorbance at 280 nm (in the tested samples), was calculated according to the procedure of Whitaker and Granum [12] and was measure using Bradford protein assay.” - Include the results (proteins) and reference of the Bradford method;

Response: The reference was added

  1. Line 423: standardize reference [22] or (Hu et al. 2020).

Response: The reference was corrected.

  1. Line 436: standardize reference [23] or (Mesa et al. 2015);

Response: The reference was corrected.

  1. Line 577: include the number of the reference;

Response: The number of the reference was included [27].

  1. In the methodology or results add the yields obtained from each extraction;

Response: We have not decided to change the methodology or results from each extractions.

  1. How much weight was used for each fruit? Include detail in methods;

Response: 1 mg of each of freeze-dried vegetables has been added in method (line 306).

  1. Fig 4B: what was the concentration used of the extracts?

Response: The concentration presented in Fig B was 50 µg/ml. This information is added in methodology in “4.7.7. Total Antioxidant Capacity of Plasma”.

  1. Remove additional information from the top of the figure (Fig 1A-E);

Response: Figures 1A-E have been corrected.

  1. Line 212: “highest content of flavonoids” - it was quantified?

Response: The lines 211-212 have been corrected ,,The zucchini preparation showed the highest radical scavenging effect of the preparations, because its phytochemical profile was the richest in terms of the presence of flavonoid compounds’’.

  1. Include references in some paragraphs of the discussion;

Response: The references was added.

  1. In vitro/in vitro: standardize word;

Response: The in vitro word was standardized.

  1. Line 241: “benzoic acid derivative level, around 260 mg on 1 g of dry” - this result was described? What about the values ​​obtained for the other samples?

Response: The sentence have been corrected (line 241-242).

  1. Line 266: remove color markup;

Response: The color markup was removed.

  1. It would be important to include the quantitative values ​​(dry weight) of the main compounds present in the samples;

Response: Only qualitative analyses were performed.

  1. I suggest performing in vitroassays on platelet aggregation. Some studies have demonstrated a relationship between oxidative stress and platelets/thrombosis (e.g.: Oxidative Stress and Platelets. Arteriosclerosis, Thrombosis, and Vascular Biology, Volume 28, Issue 3, 1 March 2008, Pages s11-s16). In addition, the antiplatelet / antithrombotic effect of polyphenols has already been demonstrated;

Response: The platelets aggregation and other antiplatelets assays will be performed and  described in next publication.

  1. Line 193: “including extracts and fractions” - which fractions?

Response: The sentence have been corrected (line 193).

  1. Suggestion: the results (figures) of the effect on blood clotting can be suppressed and the results presented only within the text.

Response: We remove the figures. The results are presented only within the text.

Kind regards

Agata Rolnik

Reviewer 2 Report

Dear authors,

Thank you for submitting the manuscript: Comparative Phytochemical, Antioxidant and Haemostatic Studies of Preparations from Selected Vegetables from Cucurbitaceae family to the Molecules.

Comments:

Lines 168-170. The antiradical capacity of the five vegetable preparations are also shown in Figure 4C. The order of the DPPH scavenging activity for the preparations was as follows: cucumber (0.007±0.00) < pumpkin (0.052±0.00) < white pattypan squash (0.064±0.01) < yellow pattypan squash (0.086±0.01) < zucchini (0.093±0.02) (Figure 4C).

Please change Figure 4C to Figure 4B.

Figure 4B. Please add Statistical analysis in Figure (*).

The manuscript needs some minor changes.

I suggest minor paper revision.

Author Response

Dear Reviewer

Thank you for heplfull comments  and suggestions on our manuscript. I would like to respond for comments below:

Comments:

Lines 168-170. The antiradical capacity of the five vegetable preparations are also shown in Figure 4C. The order of the DPPH scavenging activity for the preparations was as follows: cucumber (0.007±0.00) < pumpkin (0.052±0.00) < white pattypan squash (0.064±0.01) < yellow pattypan squash (0.086±0.01) < zucchini (0.093±0.02) (Figure 4C).

Please change Figure 4C to Figure 4B.

Response: We decided not to change the figure  4C to 4B, because figure 4B presents total antioxidant capacity of plasma and figure 4C presents the use of TLC-DPPH• assay for the detection of antioxidant activity of preparations

Figure 4B. Please add Statistical analysis in Figure (*).

Response: The statistical analysis was performed using  one-way ANOVA followed by a multicomparison Tuckey test and Kruskal-Wallis test, compared with control. There is non-significant influence on total antioxidant capacity

The manuscript needs some minor changes.

I suggest minor paper revision.

Kind regards

Agata Rolnik

Round 2

Reviewer 1 Report

Review comments:

Minor revisions/suggestions

  1. Method: Lines 290-299: please, iclude volumes used (methanol) or includ reference of method. It would be important to detail or add reference to the method of solid phase extraction (SPE).

  1. Lines 453-454 and 249: “in vivo model” - replace “in vivo model”
  2. Line: 182 ...”The destructive effect of OH. is reflected....” - replace “OH.”

Author Response

Dear Reviewer

Thank you for helpful comments  and suggestions on our manuscript. I would like to respond for comments below:

  1. Method: Lines 290-299: please, include volumes used (methanol) or includ reference of method. It would be important to detail or add reference to the method of solid phase extraction (SPE).

Respond: We corrected the method : Specialised metabolites fractions were eluted from Oasis Extraction Cartridges (Waters, Massachusetts, USA) with 6 mL of 85 % methanol.

  1. Lines 453-454 and 249: “in vivo model” - replace “in vivo model”

 Respond: We corrected the in vivo model.

  1. Line: 182 ...”The destructive effect of OH. is reflected....” - replace “OH.”

Respond: We replaced OH. with “OH.”.

Kind regards

Agata Rolnik
